# Reproductive coercion and abuse in intimate relationships: Women's perceptions of perpetrator motivations

Laura Tarzia [1,2]*, Mandy McKenzie [1]

1 Department of General Practice & Primary Care, The University of Melbourne, Carlton, Victoria, Australia,
2 Centre for Family Violence Prevention, The Royal Women's Hospital, Parkville, Victoria, Australia

* laura.tarzia@unimelb.edu.au

## Abstract

Reproductive coercion and abuse is a hidden and poorly recognised form of violence against women. It refers broadly to behaviours that interfere with or undermine a person's reproductive autonomy, specifically to promote or prevent pregnancy. Reproductive coercion and abuse can involve physical, sexual, financial or psychological abuse in order to achieve these aims, and is overwhelmingly perpetrated by men against women. As an emerging field of scholarship, conceptual understanding of reproductive coercion and abuse is still in its infancy; however, it is often described as being linked to coercive control. In this article, we seek to highlight the complexity of this relationship through qualitative analysis of in-depth interviews with 30 victim/survivors in Australia recruited from the community, focusing on their perceptions of the perpetrator's motivations. We developed four themes from our analysis: 1) His needs came first; 2) The illusion of a perfect father; 3) Creating a weapon of control; and 4) My body was his. Perceived perpetrator motivations ranged from entitlement and self-interest to a deep desire for domination and entrapment. Pregnancy preventing behaviour was more likely to be linked with entitlement and self-interest, whereas pregnancy promoting behaviour tended to be described by participants in relationships where there was a broader pattern of ongoing control and entrapment. Thus, we suggest that coercive control is a motivating factor for some, but not all men who perpetrate reproductive coercion and abuse. A greater understanding perpetrator motivations may be important for practitioners, particularly those working in sexual and reproductive health services, since it could be relevant to women's level of risk for coercive controlling behaviour or more extreme forms of physical or sexual violence.

## Introduction

Reproductive coercion and abuse (RCA) refers to behaviours that deliberately interfere with a person's reproductive autonomy [1,2]. Increasingly recognised as a major public health concern, it comprises two main forms: pregancy promoting behaviours (such as the use of threats, violence or coercion to cause pregnancy or to force someone to remain pregnant against their

**Data Availability Statement:** Data cannot be shared publicly due to the sensitive nature of the topic, and potential safety and ethical implications for participants. The de-identified data may be made available on a case-by-case basis by The

University of Melbourne's Human Research Ethics Committee (contact via robert.reid@unimelb.edu. au), for researchers who meet the criteria for access.

**Funding:** This work was funded by the Oak Foundation (https://oakfnd.org/) through the Safer Families Centre. The funding body played no role in study design, data collection, analysis, or preparation of the manuscript.

**Competing interests:** The authors have declared that no competing interests exist.

will); and pregnancy preventing behaviours (using threats, violence or coercion to force someone to terminate a pregnancy or to prevent them from getting pregnant) [3]. In both cases, RCA is primarily perpetrated against women, usually by their male intimate partners [1], though other family members can also be perpetrators [4–6]. Thus, it is most commonly understood as an interpersonal form of abuse, although we acknowledge that there is ongoing debate within the literature as to whether coercive behaviours within health systems [7], or government policies around access to abortion [8] could also be considered as forms of RCA rather than simply facilitators of it [1,9].

RCA is a relatively "new" phenomenon in that it has only received attention in research, policy and practice since 2010. To date, there remain serious inconsistencies in how it is understood and measured in research studies. As Tarzia and Hegarty have argued [1], this has led to contradictory and inconsistent findings in terms of its prevalence. Indeed, studies report rates of lifetime RCA between 8% and 30% within cohorts of women aged over 16 years, depending on the setting and how RCA was measured [10]. Risk factors are also reported inconsistently, with a lack of clarity as to whether data collection instruments are measuring RCA, intimate partner violence more broadly, or sexual assault [1]. Moreover, it is highly likely that many risk factors differ by context (e.g. high-income versus low-or-middle-income countries) [11]. There is somewhat clearer data on the outcomes of RCA, with studies suggesting that it is associated with a range of harmful impacts including unintended pregnancies [2], sexually transmitted infections [12], poor pregnacy outcomes [13] and mental health conditions such as anxiety, post-traumatic stress and depression [14–17]. As yet, however, there has been little nuance in how these findings have been reported, with RCA treated largely as an homogenous entity, with no distinction being made between pregnancy promoting or pregnancy preventing behaviours.

Within the extant literature, interpersonal RCA is also consistently linked to physical, psychological and sexual violence [3,16]. Tarzia and Hegarty, in their commentary on the conceptual dimensions of RCA, argue that these other forms of violence can be understood as the *mechanisms* by which perpetrators force women into particular reproductive outcomes [1]. In other words, whilst perpetrators can use a broad range of tactics to achieve their aims–including tampering with or denying access to contraception, rape, blackmail, financial abuse, physical violence to induce miscarriage, forced abortion and interference in medical care [18]–the shared characteristic amongst these behaviours is that they are designed to cause fear and/or impose control [1]. However, whilst this analysis is useful in terms of distinguishing RCA from other types of abusive behaviours, there is still considerable detail to unpack in understanding this hidden form of violence. In particular, the contextual differences between pregnancy preventing and pregnancy promoting behaviours are poorly understood. Similarly, little is known about how RCA might overlap with coercive control in a relationship. Defined by Evan Stark in 2007, coercive control is an ongoing pattern of physical, psychological or sexual behaviours designed to frighten, dominate, isolate, degrade, exploit and entrap a partner in an abusive relationship [19]. Whilst RCA by definition always involves some degree of control, it is unclear whether this broader pattern of entrapment described by Stark is necessarily present.

In addition to the aforementioned knowledge gaps, a critically under-researched aspect of RCA is understanding the motivations of perpetrators. Fiske [20] defines "motivations" as a product of interactions between individual factors and situational factors. Currently, little is known about what motivates men to perpetrate RCA against their partners, although Stairmand and colleagues [21] have recently reviewed and critiqued the literature on perpetrator motivations for intimate partner violence more broadly. They suggest a conceptual framework that examines motives alongside intent (which, according to Tedeschi and Felson, can be categorised as a desire to harm or a desire to obtain compliance from the victim/survivor [22]).

Stairmand et al.'s review, which considers perpetration by both men and women, identifies access to valued resources; physical or psychological safety (ie. self-defence); retributive justice, status, and deterrence as potential perpetrator motivations, contextualised by a variety of situational factors. There is also a robust psychological literature base focused on understanding people's motivations more generally. Deci and Ryan [23], for instance, developed a model that views motivation on a continuum, closely linked to self-determination. Motivation can be either extrinsic (e.g. rules, social approval, ego) or intrinsic (personal satisfaction or pleasure). Others [24,25] within the field of criminology have examined the literature relating to people's motivations for perpetrating other types of crime, however, given that perpetrators of RCA, and indeed a large portion of intimate partner violence more broadly, may not perceive their actions to be "criminal" in nature [26,27], this work is less relevant.

A further consideration is that the majority of the literature on motivation–either from the perspective of victim/survivors or perpetrators–is quantitative in nature. However, in order to better understand the nuances of RCA, robust qualitative research is needed. Currently, this evidence base is largely lacking. Although a recent qualitative meta-synthesis [5] identified 33 studies that addressed women's experiences of RCA, the majority of these studies a) did not focus on RCA behaviours as the primary phenomenon of interest; b) focused largely on identifying specific behaviours or tactics experienced by victim/survivors and c) were of mixed quality. Nonetheless, the authors did find some limited data on perceived perpetrator motivations, which included: a desire to control women's lives and restrict decision-making; ensuring compliance with the reproductive intentions of in-laws or other family members; son preference; and strictly-defined gender roles [5]. All but the first of these findings were based on limited, weak data from low-and-middle-income countries. In two studies from the US [28,29], on the other hand, African American women described their partners using pregnancy as a way of creating a long-term connection in the face of systemic inequalities such as incarceration or unemployment. More recently, a qualitative study in Bangladesh [30] also addressed women's perceptions of perpetrator motivations, finding that RCA could be perpetrated in order to solidify a new marriage or to enable divorce, because of son preference, or as a mechanism of control. Overall, however, there is very little data to help understand *why* men perpetrate RCA against their intimate partners, particularly in high-income countries (in contrast to the RCA evidence base overall, which is strongly weighted in favour of the US).

Our aim in this study is to examine these gaps in the knowledge with the goal of further advancing conceptual thinking on RCA. We focus on the context of RCA perpetrated against women by a male intimate partner, drawing on qualitative interviews with 30 victim/survivors to explore their perceptions of why RCA was perpetrated against them. We use Evan Stark's definition of coercive control described earlier to help situate the RCA behaviours described by participants within the dynamics of the broader relationship.

## Materials and methods

This qualitative study draws on Braun and Clarke's form of reflexive thematic analysis [31] to make sense of the findings. We take a sociological view of the topic, informed by a feminist framework that views men's violence against women as a product of patriarchy and inequality, whilst also recognising the influence of situational and individual factors [32,33]. We also acknowledge that violence in relationships can take different forms, not all of which are gendered, however, the types addressed in this article—RCA, sexual violence, and coercively controlling intimate partner violence–are overwhelmingly perpetrated by men against women [33,34]. We undertook this work as part of a broader project focused on understanding and responding to psychological violence in relationships.

## Study context

Our research was conducted in the state of Victoria, Australia, although participants were recruited nationally. The healthcare context is similar throughout all states and territories in Australia, with free or subsidised healthcare–including obstetric care–being provided via the taxpayer-funded Medicare health insurance scheme. Healthcare can also be provided privately at the patient's cost [35]. Australia's healthcare system is generally considered to be one of the best in the world, with a strong network of primary healthcare, specialist, allied health and nursing services [35].

Screening for family or domestic violence routinely occurs in most states across antenatal care and maternal and child health [36]. In other healthcare settings such as abortion services or pregnancy counselling, enquiring about violence comes down to the level of confidence and awareness of the individual practitioner to identify warning signs. Providers receive some limited training in how to sensitively inquire and respond, although the level of confidence and competency varies widely across services and between individual practitioners [37–40].

It is worth noting that surgical or medical abortions are legal Australia-wide (although in South Australia abortion was only legalised in 2022); however, states and territories have different legislation in regards to the timeframe when abortion can be performed and the conditions surrounding it [41]. For example, in Tasmania, surgical abortions after 16 weeks' gestation require a second medical opinion to confirm that continuing the pregnancy would be detrimental, whereas this is not required until after 24 weeks' gestation in Victoria [42]. Abortions can be provided by a range of services, including family doctors/general practitioners (medical only), gynaecologists and obstetricians, as well as in private clinics and hospitals. Medical abortions can also be obtained via telehealth services.

## Participant recruitment

A total of 30 participants were recruited to this study. Twenty-six of the participants were recruited via advertisements placed on social media by the research team. Recognising that many potential participants would not use the term RCA to describe their experiences, our advertisements asked: 'Has a partner or family member ever forced you to get pregnant, stay pregnant or end a pregnancy? If so we invite you to take part in a confidential interview about how these experiences have impacted on your relationship with your children'. Interested individuals who clicked on the advertisement were directed to a secure online form, where they were asked to provide a safe telephone number or email address (that a perpetrator could not access). The researchers then followed up with these individuals to provide a participant information sheet and written consent form, and to book in an interview time.

We recruited the remaining four participants via an online survey that we were undertaking for a separate study surveying women about experiences of psychological abuse (including RCA). The survey participants had been recruited via a research panel and had indicated their interest in being involved in further studies. We contacted those who had indicated on the survey that they had experienced RCA and invited them to take part. If they expressed interest, we provided them with a plain language information sheet and consent form. This took our total number of participants to 30.

## Data collection

The authors (XX and XX) held in-depth, unstructured interviews via telephone (n = 29) or online video conferencing (n = 1) between 8[th] September 2021 and 29[th] July 2022. The length of the interviews ranged from 23 minutes to 2 hours (average 55 minutes). We asked participants to provide us with some background information about their relationship, and to tell us

their story about what happened in the relationship in regards to pregnancy. We also asked about how these experiences might have impacted on their relationship with their child/ren (findings reported separately in a forthcoming article). The open-ended interview approach was ideal for these interviews, as it enabled participants to identify the issues that they felt were important and explain them in their own words [43]. Indeed, the majority of the interviews provided rich data about the participants' experiences. We audio recorded the interviews and had them transcribed verbatim by a professional transcription service.

## Data analysis and reflexivity

For the data analysis, we first imported the interview transcripts into the qualitative data coding program NVivo. Using a reflexive thematic approach [44], the first author read and re-read the transcripts multiple times to ensure familiarity with the dataset. The first author then coded the data descriptively, before grouping the descriptive codes together into interpretive codes that reflected a common meaning. This process was then repeated as the interpretive codes were cateogirised into overarching themes. The draft coding framework was discussed with the second author and refined, before being presented to a broader team of researchers for additional feedback. To enhance rigor, we repeatedly went back to the transcripts during the coding process to ensure that the themes were an authentic representation of the data. The first and second authors met weekly to discuss findings and consider possible interpretations.

A reflexive analytic approach recognises that themes are created through the researcher's subjective engagement with the data, rather than being 'discovered' through the analysis [44]. Considering this, we acknowledge that our interpretations of the data were informed by a feminist lens, including an understanding that both RCA and mothering are experiences that are shaped by gendered discourses constructed within a patriarchal social context. One of the authors (XX) is a sociologist who has been researching sexual violence, RCA, and their health impacts for nearly a decade. The second author (XX) is a post-doctoral researcher working in the area of violence and health with many years prior experience in the women's sector. Both are highly experienced qualitative researchers and interviewers who believe in foregrounding survivor voices to guide research, policy and practice.

## Ethical and safety issues

The study was approved by [blinded] human research ethics committee (HREC # 2021-21742-19765-3). The safety of participants and the potential for the topic to cause distress were key ethical considerations. Consistent with the trauma-informed approach to interviewing outlined by Campbell [45], our interview techniques aimed to validate participants' experiences and strengths, whilst being attuned to their emotional needs. Participant choice and control was emphasised as much as possible during the interview process. In the event of participant distress, a protocol was developed to guide responses. This protocol has been used in numerous similar studies by the authors [46,47], and involves checking in regularly with participants, engaging in supportive/active listening, providing choice around whether to continue or postpone the interview and exploring options for further support if needed.

Beyond the interview itself, we sought to maximise participant safety in a number of ways. Safe methods of communication were utilised at all times with participants (such as referring to the study as a 'women's health study' in emails, in case they were intercepted by an abusive partner). When making contact via telephone, information was only provided to the participant directly. If another person answered the phone, we did not say what the study was about. On completion of all interviews, participants were provided with a list of resources to contact should they feel distressed at a later time, and the researchers checked-in with the participant

shortly after the interview to ensure that they were not experiencing any ill-effects from the interview.

## Findings

### Participant demographics

The majority of participants recruited to this study were educated, born in Australia, and spoke English as a first language. Most were aged in their forties, although the age range overall was between 30–79 years. Just over half the participants were employed. Nine participants were currently in a relationship (with a new partner), whilst 21 were not. The majority were living with their children, and one was still living with the partner who had perpetrated RCA. Five participants described themselves as having a disability. All participants were female; twenty-nine identified as women and one participant identified as non-binary.

In terms of RCA experiences, 18 of the 30 participants were categorised by the research team as having experienced pregnancy promoting behaviour. Fourteen participants had experienced pregnancy preventing behaviour (including 3 participants who described the use of physical violence during pregnancy which they interpreted as being intended to cause a miscarriage). Five participants had experienced both pregnancy preventing and pregnancy promoting behaviour by the same perpetrator. All the participants' experiences of RCA were with a male intimate partner, although a few had also experienced co-occurring coercive behaviours from another family member.

There were differences amongst participants in terms of whether physical or psychological abuse or coercive control pre-dated the RCA. For some women, pregnancy was forced or coerced as part of an ongoing pattern of abuse, whilst for others, the pregnancy was the trigger for abusive behaviour to commence for the first time.

### Themes

We developed four themes from our reflexive thematic analysis of the interview data, describing the motivations of perpetrators as perceived by the participants in our study. These themes are: 1) His needs came first; 2) The illusion of a perfect father; 3) Creating a weapon of control; and 4) My body was his. Each theme has been outlined in detail below with supporting quotes from participants. Where possible, we have reported the quotes verbatim, with minor corrections only to enhance clarity.

**His needs came first.**   This theme describes the experience of participants who felt that their partners were primarily motivated by self-interest and entitlement when perpetrating RCA, rather than it being part of a broader pattern of fear and control in the relationship. These participants almost all experienced RCA in the form of pregnancy preventing behaviour, where a partner pressured, forced or frightened them into having an abortion. Their stories overwhelmingly focused on an accidental pregnancy, often occurring because they were unable (due to health issues) to use more reliable methods of contraception. Thus, the pregnancy itself was usually not coerced; the RCA behaviours occurred *after* the pregnancy was disclosed. For example, one participant described how, when she refused to acquiesce to her partner's demands to terminate her pregnancy, he stopped speaking to her:

> My ex-husband didn't speak to me for three years. . . He's wanted nothing to do with the child. It was heartbreaking. . . He would just say the necessaries, sort of thing, but he dropped helping in the house. He was as silently unpleasant as he could. (Participant 27)

Another woman described her partner's initial reaction when her unplanned pregnancy was revealed:

> That morning, it was like a switch flicked, it was like a totally different person. Even things like the tone of voice changed, it was like I had done this one horrible thing to him. I mean, I was on birth control, he'd been at the doctors with me where I got the birth control, it just failed. (Participant 24)

The above participant articulates clearly how her partner framed the unplanned pregnancy as something she had *done to him*; centering himself in the narrative and behaving like a victim. Another participant described how, when discussing what they should do about the pregnancy, her partner refused to consider any alternatives to a termination, due to his perception of what "people would think" of him:

> I remember saying one morning to my boyfriend at the time, he was—we were sitting at a café, and I said 'Well what about adoption? We could think about that?' He goes, 'No, no, no, I do not want you walking around pregnant, people will see you pregnant and I don't want that. . .People will look at me, people will think I'm a loser for being a deadbeat dad.' (Participant 1)

Whilst some participants admitted to being ambivalent about the pregnancy, this did not necessarily mean that they wanted to have an abortion. Many wanted the time and space to consider their options before reaching a decision. Yet, their partners refused to consider anyone's needs except their own, using a variety of emotional and verbal tactics to ensure that their wishes were complied with.

> I didn't particularly want to have baby, but I also didn't particularly want to have a termination either. I wanted to have the opportunity to talk about the possibility of having another baby. He was quite a prick about it, pretty horrible about it and really verbally abusive towards me, until I told him that I had decided to have a termination.(Participant 10)

Interestingly, many participants described their partners as being dismissive of the "problem" of the unplanned pregnancy, despite being highly invested in the outcome. The men seemed to perceive that having a termination was "no big deal" and that women were being unecessarily emotional about the situation. This led to an incredibly callous and uncaring attitude from many of the perpetrators, with participants recounting being dropped off at the abortion clinic or left at home alone to take an abortifactant medication, with no support or concern for their wellbeing, just a demand that they "deal with it".

> He picked me up [from the abortion clinic] and took me back to my house and dropped me off. I was still groggy and not well and my boys were at home. And he said he had to go. He just left me there, and instantly I had to start cooking them dinner and, oh it was just horrible. . . If I'd get upset or sad about it, it was like, you're not allowed to talk, don't talk about it, look at what you're doing, you're just, you know, trying to guilt me and trying to make me feel bad. (Participant 4)

What is critical to note about the perpetrators in this group, is that the vast majority eventually left the relationship, even after the participant had complied with his wishes. This highlights that–whilst perpetrators may indeed exhibit behaviours that are intended to control the

outcome of a specific pregnancy–broader control or entrapment within the relationship is not necessarily their aim. As the below participant articulates, often the perpetrator's actions were perceived as coming from a place of selfishness rather than a need to dominate:

> He didn't want to be tied down, he didn't want the maintenance if I had the baby, he wanted none of the entrapment, he just wanted to be a golf professional and travel and things like that. (Participant 19)

**The illusion of the perfect father.** In the second theme developed from our analysis, participants described their partners as being motivated by an obsessive attachment to the illusion of a perfect family, and the idea of themselves as a perfect father. For these women, pregnancy happened as a direct result of the perpetrator's actions (RCA in the form of pregnancy promoting behaviours). Some perpetrators tampered with, or otherwise sabotaged the participant's contraceptive choices, some undermined their sexual autonomy, and others pressured, badgered or emotionally blackmailed them into "giving in" and agreeing to try and fall pregnant. Many participants described their partners as having strong views about the timing of pregnancy, the number of children and the spacing between them. As a result, they felt that their own needs and wishes were irrelevant; their role was simply to help facilitate the perpetrator's fantasy.

> I think I was just like a workhorse to provide him with children to adore him. It felt a lot like that. (Participant 9)

> He used me for reproduction. . .not to have a loving, committed relationship with, but because he wanted children. (Participant 20)

Some participants perceived that the perpetrator wanted a child in order to boost his image and credibility. Several spoke about their partner's need to look good in front of friends, family members, or broader society, believing that having a baby would help achieve this. For others, successfully impregnating a partner was perceived as giving a boost to his masculinity by being an outward sign of virility.

> He felt he didn't get taken seriously enough in life by his peers and if he was to have a child, it would, yeah, he'd be taken more seriously as well because he would be, like, more normal. (Participant 4)

> I felt like he showed me off a lot to people. He was always wanting me to go places and I really didn't want to do it. It was all about showing me to everyone; family members, friends, friends he hadn't seen in years. Yeah, showing me off to people that I was pregnant. (Participant 5)

> It's a real alpha male kind of dominance thing, 'I have all of these children which makes me more of a man than you because I can father these kids, which means I've got the bigger dick or bigger balls.' (Participant 7)

Yet, after successfully coercing their partner into having a child, many of these men wanted very little to do with the reality of parenting, pregnancy and childbirth. Most were extremely hostile to the idea of changing nappies, feeding and caring for an infant, or undertaking any extra household duties.

He wouldn't help with feeding them, he wouldn't help with getting a bottle, he wouldn't help with getting me to rest during the pregnancy. I had pre-eclampsia, but I was expected to go and do the mowing. He wouldn't help with that. (Participant 15)

This disinterest led many of the participants to describe what we have chosen to refer to as "performative fatherhood"; a father who looks good in public, but in private is disinterested at best, and abusive at worst.

He would hold her [the baby] in public so everyone could goo and gah over him and praise him for being such a great dad. But when it came down to the crunch at home, he wasn't interested in actually doing any of the not-so-fun work stuff. (Participant 11)

You know doing the night sleeping thing, when you try and get them to sleep through the night? So, I came back from hospital–I was really sick. . . He told everyone how he was up during the night looking after [the baby]. . .He'd tell everyone how he fed her and did all this. Instead, he was watching porn and chatting to women [laughs]. (Participant 26)

As their children grew older, many participants recounted the perpetrator's need to ensure that all members of the household complied with his vision of the perfect family. This could include policing behaviour or using emotionally abusive tactics to ensure that the family was run the way he wanted it to. As one woman articulated, they were treated like possessions to be rolled out to boost the perpetrator's public image; with little regard for their own needs and, again, often in direct contrast to his private behaviour:

He was an artist and we were trotted out when he was going to exhibitions. We were told we had to come and support him and show ourselves. There was this public image, basically, of the mother and the children and the supportive family. How successful he was at home and in the community. How he's a nice guy. Versus the way he would behave at home. (Participant 15)

The pressure to to conform to the perpetrator's expectations was strong for many of the participants, with their partners punishing them for behaviour that did not align with their vision.

I think his focus on his kids was a bit like they were items on a stock exchange, and if they didn't perform, love was conditional on performance and control. In a way, I mean if you think about it, it's the same as my relationship with him. If I didn't perform in the way that he deemed, then love wasn't forthcoming so to speak. (Participant 22)

Another participant, whose first child was extremely clingy and unwilling to be held by anyone other than his mother, articulated how her partner accused her of deliberately manipulating their child. His abusive behaviour only ceased when she agreed to have a second child, which would be "his baby":

He felt that I had somehow orchestrated this, that I had turned our baby into *my* baby and our baby's sort of unwillingness to be looked after by him or anyone else was somehow my doing; that was all in my control and that I had sort of messed up the first baby, so he was always applying pressure on me to have another one. The first one ended up being for me; the second one would be for him. . . [Second baby] was really fearless and social. So her dad got to interact with her a lot more and the extended family as well. So he was happy. He got

—he finally got to have what he wanted out of a baby, a baby that met his needs and his expectations. (Participant 3)

Similar to the first theme, for some participants in this group the perpetrator's behaviour was within the realm of entitlement and self-interest rather than coercive control. For example, when women took their children and left the abusive relationship, the perpetrator often quickly set himself up with a new partner with whom he sought to create another "perfect family", rather than seeking to maintain control over the first one. On the other hand, for other participants, perpetrators enacted a pattern of controlling and dominating behaviour when seeking compliance with their vision of the perfect family–often with the use of tactics that elicited fear.

**Creating a weapon of of control.**   The third theme from our analysis very clearly aligns with severe coercive control in relationships. Participants described how their ex-partners had deliberately sought to impregnate them, knowing that a child was the ultimate "tool" or "weapon" for maintaining control in the relationship. Many participants believed that the perpetrator saw their children not as human beings but as pawns to be used as a means to an end:

He saw that having a child gave him a massive amount of control. I don't know if that's what it is, because it's not that he loves kids. (Participant 17)

I'd say he never wanted the children. He didn't want the responsibility. The children were his pawns to control me. (Participant 18)

This control often started before a child was even born–with perpetrators exploiting the vulnerability that came with pregnancy–and continued as the baby grew older.

The manipulation, the coercive control was already there. . .and once I was pregnant and vomiting, I was so vulnerable. So vulnerable. (Participant 9)

You can control your spouse if you know that she's stuck at home with a baby. If you know that she's got a baby, or she's got a big, fat pregnant belly. . . If she's stuck at home with a baby she can't go anywhere, she can't do anything, she can't work, she can't achieve things for herself, because babies are hard work and require an awful lot of time and dedication. (Participant 11)

Some participants described how their ex-partners would repeatedly use their child as a way to coerce them into doing things against their will:

He'd [the baby] be sleeping in the port-a-cot in the room. . .in the bassinet or whatever. We'd have an argument and then if the baby would wake up from the noise—I'd obviously go to the baby to pick him up and my ex would block me from going to him. So it was like the baby just became the ultimate tool to coerce me into doing things I didn't want to do or get back at me when he was angry at me. (Participant 5)

He used her–if he really wanted me to do something I didn't want to do, it was 'oh, [daughter] will be hurt'. (Participant 26)

Children were often used as a mechanism to trap women in the abusive relationship, but even once the relationship had ended, perpetrators still used children as a way to insert themselves back into participants' lives. One participant described how her ex-partner would insist

on lengthy FaceTime conversations with their two-year-old child, even though the child was too young to meaningfully participate:

> He wanted the calls to go for 20, 30 minutes. He just would not hang up the phone. My son wasn't even really talking at that stage. . .and it felt like having him [the perpetrator] in my house. He knew the calls were making me uncomfortable because I started saying, 'Hey, do you think we could drop the time back or do you think we could drop the frequency back or whatever?' He was like, 'No, I'll be calling every day. He'll get used to it'. . .[I felt like he was] forcing his way into my home. (Participant 5)

Others spoke of their frustration and shock when their ex-partners–who had previously shown no interest in their children–initiated protracted legal proceedings against them after they separated. Although these legal proceedings were ostensibly to obtain custody of the children, participants felt that it was simply a way to maintain contact with them:

> I was so naïve and I talk to so many parents who have no clue–they're naïve to this too–the power a man has over a woman who has his child. I had no idea. I thought, 'Cool, I'll just bring up this baby all on my own because he doesn't want anything to do with it'. [I didn't realise] that he can just change his mind, get lawyers involved and ruin my life, basically. . . Did he know, secretly, that he was now going to have this control over me because I'd dared to leave him, I'd dared to call out his behaviour and say 'This isn't OK' and he's not used to that? (Participant 6)

> Whenever [my son] is with me, I'm fighting a court case. . .It was kind of like a plan to control [my] life. . . Here [in Australia], until my baby turns 18, I have to deal with [him]. He's got access to me. It's very terrifying for me. (Participant 12)

**My body was his.** The final theme from our analysis describes the most extreme experiences of RCA in the context of extensive and ongoing coercive control. Participants whose experiences aligned with this theme described either pregnancy promoting behaviour alone, or–for five of the participants–both pregnancy promoting and pregnancy preventing behaviour from the same perpetrator. In this theme, RCA was an extension of the perpetrator's domination and control over the participants' everyday lives, and represented an escalation of these behaviours. This experience is different to the second theme ("The illusion of the perfect father"), where participants described being used as incubators to fulfil the perpetrator's needs. In this final theme, the motivation of the perpetrator went beyond using their bodies for his own reproductive purposes; rather, women's bodies were used as a mechanism for maintaining psychological domination and control.

Several participants described having very little bodily autonomy in their relationships. Their partners frequently made comments about what they should wear and how they should look. One woman articulated how her partner would claim ownership over her body, not even allowing her to shower alone. This feeling of entitlement to her body only increased during pregnancy:

> It was really scary. I wasn't even allowed to say no to him touching me, or like have a shower with the door closed, because he had to watch me have a shower, and he would say 'No, your body is mine, I can do what I want with it. You are just a vessel for this child'. (Participant 16)

Related to this, participants in many cases also lacked sexual autonomy and control over their contraceptive choices. Perpetrators used verbal coercion, emotional blackmail, and physical violence to force women to make themselves sexually available, whilst concurrently not being allowed to use contraception.

> I had no control, no control. It was nothing for me to wake up in the middle of the night with him on top of me. In the end with the children, if he didn't get what he wanted, the children copped his temper the next day. So, I had to give him what he wanted to protect the children. (Participant 18)

Participants were expected to have sex with their partners even after having miscarriages or ectopic pregnancies, or within days of giving birth.

> I had an ectopic pregnancy. . .and got carted off to hospital. . . As well as removing the embryo in my tubes, I then had to go and have a termination as well. I came out of that feeling pretty bad, and he fairly quickly, I think the day that I came home, wanted sex again. (Participant 23)

Several of the participants gave thoughtful reflections on why they thought their partners chose to attack their reproductive autonomy in particular. Some felt that a woman's control over her own reproductive choices was like the "last bastion" of autonomy; even if control over other aspects of life were taken away, at least they were still in charge of what they did with their bodies. For those participants who experienced both forced pregnancy and forced abortion, this was particularly salient, because it was evident that the perpetrator did not care about the outcome of the RCA. Rather, what motivated them was the sense of power associated with being able to exert reproductive control over a woman.

> It was a control thing. 'I can get you pregnant, now I can make you not have it.' (Participant 10)

> He coerced me into having another baby and for the first three months of my pregnancy I was really sick with morning sickness and he still forced himself onto me daily. . .I almost lost the baby after one event before Christmas, and at 12 weeks old, 12 weeks pregnant, I thought I was going to die. . .The next weekend I got out and he tried to make me have an abortion. That's the kind of control that he wanted to have. He could make me do anything and he said, 'I will make you do anything I want you to do.' (Participant 16)

Connected to this, one participant believed that her ex-partner used RCA to "break her down" psychologically. She recounted how she had been a strong, successful woman prior to meeting her partner, and described how through a combination of RCA and other forms of violence he gradually eroded her confidence, her career and her mental health.

> They [perpetrators] don't want submissive women; they want strong women that they can break down. I think reproductive choice is–out of everything, that's probably the one thing that I have. . . Maybe for men, that might be the strongest thing for them to break down on a woman because that's I think one thing that women try and hold on to as much as possible. . . I think once a guy breaks down your reproductive control, then they know that they've pretty much won. (Participant 26)

**Summary of findings.** Four themes were developed from women's accounts of RCA, describing four contexts and motivations for this hidden form of abuse: 1) His needs came

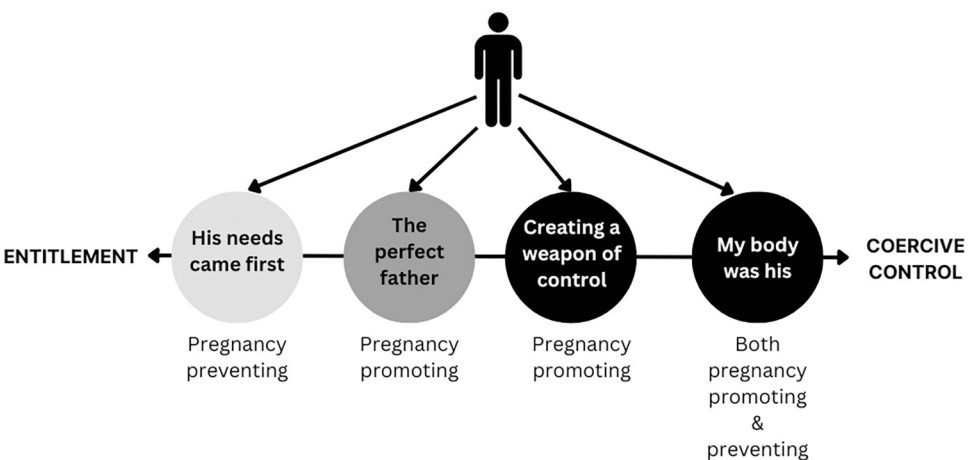

**Fig 1. Types of RCA behaviours and perpetrator motivations.**

first; 2) The illusion of a perfect father; 3) Creating a weapon of control; and 4) My body was his. This analysis suggests that the motivations of perpetrators may fall broadly into two camps: those who are motivated by a sense of aggressive self-interest and entitlement, and those for whom RCA is perpetrated as part of a broader context of coercive control. This is illustrated in Fig 1 below.

Pregnancy preventing behaviour was more likely to be linked with entitlement and self-interest, whereas pregnancy promoting behaviour tended to be described by participants in relationships where there was a broader pattern of ongoing control and entrapment.

## Discussion

RCA is often described as being a "type of coercive control" [48,49], and indeed, fear and control are at its core [1]. However, it is important to keep in mind that just because a behaviour is *controlling* or *coercive* does not make it "coercive control" in the sense that Evan Stark intended the term to be used [19]. Here, we specifically refer to coercive control as an ongoing pattern of physical, psychological or sexual behaviour designed to subjugate, instill fear and entrap the victim/survivor in the relationship. Whilst our findings show that RCA is certainly perpetrated in this context by some men, there are also other perceived motivations and contexts that are important to explore and understand. In short, RCA is far more diverse and complex than what the evidence has previously suggested.

A key finding of our study is that some perpetrators of RCA may be motivated by a sense of aggressive self-interest and entitlement rather than a desire to exert dominance and control over a partner within the relationship. This does not mean that the behaviour of the perpetrator was not abusive or harmful, or that their tactics do not represent forms of emotional/psychological violence, and in some cases, physical violence. Neither does it mean that participants did not experience fear of negative repercussions if they did not comply. Rather, the findings highlight that the perpetrator's desire for control was limited to a specific reproductive outcome rather than being a pattern of broader control and entrapment in the relationship. This is consistent with the finding that many of these men perpetrated pregnancy *preventing* behaviour, attempting to avoid being "tied down" by the responsibility of a child. Moreover, perpetrators often left the relationship, even after achieving the desired reproductive outcome. Consequently, it may be that the violence perpetrated by these men is more in-line with what Johnson, in his analysis of perpetrator typologies, has described as "situational

couple violence" [50], which is violent behaviour that is limited to a particular stressor or context. It is important to note, however, that whilst situational couple violence in Johnson's original typology is sometimes described as being mutual [50], according to the participants in this study, this was not the case in their relationships.

Similarly, for some (but not all) men motivated by the illusion of the perfect father, it was apparent that whilst their own sense of entitlement to the "perfect family" was a driving factor in their perpetration of RCA, they had little desire to be involved in the realities of everyday parenting. Buchanan and Humphreys [49] reported a similar finding of partner disinterest in their qualitative study of women's experiences of coercive control during pregnancy, birthing and postpartum; however, they interpret this behaviour as a deliberate attempt to hurt the woman. In our study, on the other hand, some participants suggested that having children was about a male partner acquiring possessions to enhance their community standing or ameliorate insecurities about their masculinity rather than being tools of control. Women's bodies were used as vessels for achieving this particular reproductive aim. Having said this, a subset of men did exhibit elements of coercive control in their aggressive policing of the family unit, expecting that women and children comply with their fantasy or risk repercussions. This finding is consistent with much of the literature on mothering in the context of violence [49,51], but has been underexplored in the context of RCA.

Along the continuum of coercively controlling behaviour, participants who felt that RCA was deliberately perpetrated in order to use children as a weapon of control, or for whom RCA was an extension of the partner's erosion of their bodily and sexual autonomy, clearly described overlaps with the most extensive and severe coercive control. There is a growing body of literature highlighting how perpetrators of coercive control use women's children against them, both during, and after an abusive relationship has ended [52,53]. Our findings suggest that some perpetrators planned for this in advance, knowing that having a child would increase their level of control. Participants who had experienced RCA in both forms by the same perpetrator also seemed to be at high risk of ongoing psychological, sexual and physical violence. Perpetrators in these instances used women's reproductive capacity in order to dehumanise, degrade, and dominate them psychologically. This behaviour is clearly gendered, and consistent with recent scholarship on intimate partner sexual violence [18,46,54], where women describe similar attitudes of bodily ownership on behalf of male perpetrators. Numerous scholars [34,52,55] have written about how perpetrators of coercive control seek to weaponise women's social roles (as mothers, as sexual partners, as homemakers) against them as a way of eroding their self worth. Our study extends this one step further to highlight how perpetrators of RCA weaponise women's physical bodies and reproductive organs to reinforce dominance and maintain control.

Taken together, our findings build on our previous suggestion that perpetrator intent is a critical factor in understanding RCA. Based on the voices of victim/survivors in this article, we argue that intent is not only central to distinguishing RCA from other types of abusive behaviours in relationships (e.g. intimate partner violence, sexual violence) but is also critical to making sense of the different *types of RCA* and how they intersect with coercive control or function as indicators of possible risk of harm. A key finding of this study is that pregnancy preventing behaviour may not happen in the context of coercive control for some women, but rather, stems from a sense of decision-making entitlement and selfishness on the part of the perpetrator. Pregnancy promoting behaviour, on the other hand, may be more closely linked to coercive control in the traditional sense. Douglas and Kerr [18] have also explored links between forced pregnancy and coercive control in their study of victim/survivor experiences with the Australian legal system, reaching a similar conclusion. Given that many commonly-used measurement tools for RCA only ask about pregnancy-promoting behaviours [56,57],

this may be why RCA is frequently associated with homicide risk, stalking, traumatic brain injury, and other physically harmful forms of intimate partner violence in epidemiological studies [16]. Yet, our findings suggest that this does not paint a complete portrait of a woman's level of risk. In light of this, we argue that measurement tools need to incorporate questions about pregnancy-preventing behaviours, as well as considering situations where pregnancy promoting and pregnancy preventing behaviours are perpetrated by the same person.

Similar distinctions can be seen when loosely applying motivation frameworks such as those mentioned earlier, although we did not use these frameworks to shape our analysis. Perpetrators in the first two themes could be viewed as having extrinsic motivation, since they seemed to be largely influenced by their concerns about their social status and what they thought others would think of them, whereas those in the third and fourth themes seemed to be more intrinsically motivated by their own personal satisfaction in having power over their partner.

Although our findings present a more complex and nuanced understanding of RCA than what has been offered to date, it is important to consider that we are not proposing a strict typology of perpetrator types or suggesting that any of the patterns we have identified are definitive. Some perpetrators displayed different motivations at different times, and the patterns of experience described by the participants were not always mutually exclusive. Neither are we suggesting that all forms of RCA–irrespective of perpetrator intent–do not have the potential to cause serious harm to victim/survivors; on the contrary, research overwhelmingly suggests the opposite. Rather, we wish to highlight the diversity of perpetrator motivations, putting forward the theory that there is a subset of RCA perpetrators for whom establishing and maintaining coercive control is not the ultimate goal, even though they may be behaving in ways that are abusive and violent. Furthermore, even amongst perpetrators whose overall motivation *is* coercive control, the impetus driving these behaviours can differ. We suggest that more research is needed in this area to drive a greater understanding of RCA. In particular, our model of perpetrator intent could be further explored in the context of RCA perpetrated by other family members, and in diverse cultural contexts, to see whether it can be mapped on to women's experiences.

## Implications for practice

Our findings have important implications for practice, particularly in the context of abortion and antenatal care, and for men's behaviour change programs. These are outlined below:

1. Practitioners who suspect or identify that patients are experiencing RCA could seek to ascertain the woman's perception of the perpetrator's motivation when enquiring sensitively about their experiences. Understanding this contextual detail may help to determine her level of risk of being subjected to ongoing coercive control. Although women may not always know or understand the perpetrator's reasoning, many are able to make an "educated guess" as they have first-hand and extensive knowledge of the perpetrator.

2. Women who are being forced or pressured to have a termination against their will may not always be experiencing coercive control or be fearful for their safety; however, neither can coercive control be discounted for these women.

3. Some women experience *both* pregnancy promoting and pregnancy preventing behaviour from the same partner, often reporting very severe controlling behaviours, physical and sexual violence.

4. Women experiencing pregnancy promoting behaviours may be at elevated risk of coercive control depending on the motivation of the perpetrator. Our findings suggest that having

low or no bodily or sexual autonomy, or a partner who treats children as tools of control should be treated as red flags by practitioners for more severe, controlling abuse and violence.

5. Practitioners who work with men individually in counselling or behaviour change programs (or batterer interventions as they are called elsewhere) may find it useful to target particular attitudes or beliefs in their therapeutic work. Men who disclose that they have tried to control a partner's reproductive choices could be questioned as to what drives this behaviour.

## Limitations

Our study has several limitations. First, only women who had experienced RCA from a male intimate partner were recruited. Thus, our findings are limited to RCA ocurring in the context of an intimate relationship and do not speak to the possible typologies of RCA perpetrated by family members. This is a critical area for exploration, since research suggests that women from culturally and linguistically diverse backgrounds [4,6] and women with disabilities [58] are particularly likely to be subjected to these forms of RCA.

Second, our analysis relies solely on the perceptions of victim/survivors rather than exploring the views of men who have perpetrated RCA. Whilst it is vital that the voices and experiences of victim/survivors be foregrounded, they are only able to guess at what their partners or ex-partners' motivations were. Given the complexity of human motivation and the varying influences of psychological, social, and situational factors, it is unlikely that victim/survivors have insight into the entire picture. Moreover, they can only reflect on their experiences through a subjective lens and a perspective that is likely to be shaped by emotions such as fear, anger or distress. Consequently, obtaining men's views on why they perpetrate RCA is a critical missing piece of the puzzle that could help to understand RCA still further, and we recommend this as an area for further research.

Third, and most importantly, the experiences of largely white, educated women living in a first-world country such as Australia are likely to be vastly different to women living in other cultural and socio-political contexts [11]. The motivations of perpetrators are also likely to differ greatly by setting [30]. Thus, we do not make any claims as to the global applicability of our findings. Nonetheless, we suggest that they are a useful starting point from which to explore how the motivations of perpetrators may differ, and how these can contribute to overall assessments of women's safety in relationships.

## Conclusion

Whilst we have argued elsewhere that the central defining elements of RCA are fear and control, examining women's experiences in detail allows for a more nuanced picture of how the manifestation of fear and control can differ by behaviour and context. Our findings have important implications for measurement of RCA, as well as for practitioners working across a range of health and specialist services, including those targeting male perpetrators.

## Acknowledgments

We are indebted to the women to gave up their time to participate in this research and share their experiences. Without their voices, this work would not have been possible.

## Author Contributions

**Conceptualization:** Laura Tarzia.

**Data curation:** Laura Tarzia, Mandy McKenzie.

**Formal analysis:** Laura Tarzia, Mandy McKenzie.

**Funding acquisition:** Laura Tarzia.

**Investigation:** Laura Tarzia, Mandy McKenzie.

**Methodology:** Laura Tarzia, Mandy McKenzie.

**Project administration:** Laura Tarzia, Mandy McKenzie.

**Writing – original draft:** Laura Tarzia.

**Writing – review & editing:** Laura Tarzia, Mandy McKenzie.

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
