## [Decision Letter · Decision Letter 0]

3 Jan 2024

PONE-D-23-29096Reproductive coercion and abuse in intimate relationships: Understanding perpetrator motivations and overlaps with coercive controlPLOS ONE

Dear Dr. Tarzia,

Thank you for submitting your manuscript to PLOS ONE. After careful consideration, we feel that it has merit but does not fully meet PLOS ONE’s publication criteria as it currently stands. Therefore, we invite you to submit a revised version of the manuscript that addresses the points raised during the review process.

Although the reviewers noted significant strengths in your manuscript, they also requested attention to several issues before its publication. My own review of the manuscript leads me to agree with the thrust of their comments.  Therefore, I invite you to prepare a revised version of the paper.

We look forward to receiving your revised manuscript.

Kind regards,

Michal Mahat-Shamir, Ph.D.

Academic Editor

PLOS ONE

“The authors gratefully acknowledge funding from the Oak Foundation to support this work. We are also indebted to the women to gave up their time to participate in this research and share their experiences. Without their voices, this work would not have been possible.”

“This work was funded by the Oak Foundation (https://oakfnd.org/) through the Safer Families Centre. The funding body played no role in study design, data collection, analysis, or preparation of the manuscript.”

4. In the online submission form you indicate that your data is not available for proprietary reasons and have provided a contact point for accessing this data. Please note that your current contact point is a co-author on this manuscript. According to our Data Policy, the contact point must not be an author on the manuscript and must be an institutional contact, ideally not an individual. Please revise your data statement to a non-author institutional point of contact, such as a data access or ethics committee, and send this to us via return email. Please also include contact information for the third party organization, and please include the full citation of where the data can be found.

Reviewers' comments:

Reviewer's Responses to Questions

**Comments to the Author**

1. Is the manuscript technically sound, and do the data support the conclusions?

Reviewer #1: Yes

Reviewer #2: Yes

2. Has the statistical analysis been performed appropriately and rigorously? 

Reviewer #1: N/A

Reviewer #2: N/A

3. Have the authors made all data underlying the findings in their manuscript fully available?

Reviewer #1: Yes

Reviewer #2: No

4. Is the manuscript presented in an intelligible fashion and written in standard English?

Reviewer #1: Yes

Reviewer #2: Yes

5. Review Comments to the Author

Reviewer #1: The article "Reproductive coercion and abuse in intimate relationships: Understanding perpetrator motivations and overlaps with coercive control" provides an exceptionally enjoyable read, skillfully combining clarity and depth in its exploration of the delicate subject of reproductive coercion within intimate relationships. The author demonstrates an outstanding mastery of the topic, captivating the reader from the opening paragraphs through clear presentation and compelling argumentation. The study's focus on understanding the motivations of perpetrators and their connections to coercive control holds significant interest in the context of domestic violence. Furthermore, the methodology employed in this research stands out for its rigor, offering a systematic approach that enhances the credibility of the presented conclusions. This combination of smooth writing, a compelling subject, and high-quality methodology makes this article a noteworthy contribution to the understanding and awareness of reproductive coercion and its intersections with coercive control in intimate relationships.

Notwithstanding the overall positive evaluation, I have a few comments and reservations regarding this study.

The introduction is intriguing but revolves solely around the subject of study, overlooking the concept of motivation. However, numerous criminological theories address the issue of criminal motivation. It is important to note that several studies indicate that perpetrators of intimate partner violence are generalists rather than a distinct criminal category. A recent synthesis by Koegl & Farrington (2022) highlights 17 motivations. In his situational action theory, Wikström (2014) suggests that a crime is the chosen action of an individual in response to a specific situational motivation. Motivation is conceptualized as the result of the interaction between the individual (preferences, sensitivities, and commitments) and the situation (opportunities, constraints).

Deci & Ryan (2002) identify different motivational orientations associated with varying levels of self-determination, offering a useful conceptual framework for understanding the studied phenomenon. Intrinsic motivation refers to engagement in an activity for the inherent pleasure and satisfaction derived from it, resulting in high self-determination. Extrinsic motivation occurs when engagement in a behavior depends on instrumental reasons, indicating lower self-determination. Finally, the construct of amotivation refers to the absence of motivation and represents the lowest level of self-determination.

In my opinion, it would be crucial to review the conceptualization of the motivation concept.

The theoretical framework (ecological feminist framework) should be presented in greater detail (proposals, principles).

I'm curious to know if there's any information available regarding the victimization trajectories of these women within the context of intimate relationships, especially prior to the incidents of (RCA). This includes details on the forms of violence, duration, and any relevant information. Specifically, I'm interested in understanding whether the dynamics of violence have undergone any changes during pregnancy.

Examining the motivation of a perpetrator of domestic violence solely based on his partner's perception may introduce several potential challenges and limitations to the research. Here are some considerations:

1. Subjectivity and Bias: Relying solely on the victim's perspective can introduce subjectivity and potential bias into the analysis. The victim may have a subjective interpretation of the perpetrator's motivations, influenced by emotional distress, fear, or personal experiences.

2. Incomplete Picture: The victim may not have access to the complete range of the perpetrator's motivations. Certain factors or aspects of the perpetrator's psychology, background, or external stressors might not be fully known or understood by the victim.

3. Lack of Cross-Verification: Relying solely on one perspective limits the ability to cross-verify information. Including multiple perspectives, such as the perpetrator's own account or insights from other sources, can enhance the reliability and validity of the findings.

4. Complexity of Motivation: Understanding human motivation is inherently complex. Motivations for abusive behavior can be multifaceted and influenced by a range of personal, psychological, social, and situational factors. A comprehensive analysis requires a nuanced and multifactorial approach.

To address these challenges, a more comprehensive research approach might involve triangulating information from multiple sources, such as interviews with both partners, observations, and perhaps external records or reports. This holistic approach can provide a more nuanced and accurate understanding of the motivations behind domestic violence. In my opinion, the part in the limits in the discussion (2nd limit) that deals with this should be improved.

Reviewer #2: Review: Reproductive coercion and abuse in intimate relationships: Understanding perpetrator motivations and overlaps with coercive control

I commend the authors on an excellent manuscript. Below please find my suggestions for minor revisions prior to publication.

1. I find the title overly long. I suggest removing “and overlaps with coercive control.” This phrase is confusing for the reader who is not an expert in IPV literature.

2. I think the themes should be included in the abstract.

3.

4. Pg. 3, line 54: RCA is “conceptually contested.” Please explain how and why.

5. Pg. 3, line 57: RCA rates of 8% and 30%. Percent of what? All adult women (where)? Mothers? Women who experience IPV?...

6. Pg. 3, line 60: How does national income level affect risk factors?

7. Pg. 5, line 102: There seems to be a contradiction here. The authors state that, “there is little data regarding motivations for RCA, particularly in high income countries.” They then point out that the “evidence base overall is strongly weighted in favor of U.S. and similar settings.” Isn’t the U.S. a high income country? Perhaps the order of the two sentences should be reversed. Also, what is a setting “similar to the U.S.”?

8. Pg. 7, line 147: I understand that the findings presented here are part of a larger study that also dealt with the perceived impact of RCA on mothers’ relationships with their children. Were all of the participants mothers? What about women whose partners pregnancy prevention practices left them without children?

9. Please state that none of the women was currently living with the partner who had been abusive.

10. Pg. 7, line 162: How many of the interviews were conducted in on-line video conferencing and how many in telephone calls? Why not do video conferencing for all interviews? The authors state that there were 30 participants and 33 interviews. Were three participants interviewed twice? If so, why? Where three interviews/participants excluded from the study? If so, why?

11. Pg. 9, line 202: “In the event of participant distress, a standard protocol was developed to guide responses.” Please provide details of this protocol and how it was developed.

12. Pg. 10, line 213 : What does “primarily educated” mean?

13. The themes are well articulated. The findings section presents an analysis that is well-balanced between description and interpretation, leading to conceptual categories that are further developed in the discussion.

14. I’m not sure that the title of the third theme captures its essence. How about, “A real man must create and illusion of the perfect father.” Also, at some point (either in the introduction, at the beginning of the findings section, or at the beginning of this theme) you should note that RCA comprises not only fathering a child, but also of the way one enacts fatherhood. This is apparent in the third and fourth themes.

15. I suggest adding a summarizing sentence at the end of the findings section.

16. There is a nice integration between the findings and the existing literature in the discussion.

17. Pg. 23, line 556: Not all of the readers are familiar with Evan Stark and his writings. If you wish to highlight him, situate him within the literature on IPV and cite his definition of coercive control (consider placing it in the introduction of the article) or make it clear that you are using his definition as a working definition for your research.

18. Pg. 25, line 584: “The situational couple violence in the study was not mutual.” Add “according to the participants.” Were the women aske about their use of coercion in the relationship?

19. Pg. 621, line 621: “IPV, SV” Abbreviations should not be used the first time a term is mention.

20. Pg. 29, line 684: Change “interrogating” (which is a criminal term) to “exploring” (which is a qualitative research term).

One final note: The manuscript was very well written. It was a pleasure to read.

6. PLOS authors have the option to publish the peer review history of their article (what does this mean?). If published, this will include your full peer review and any attached files.

Reviewer #1: No

Reviewer #2: **Yes: **Chaya Possick

---

## [Author Response · Author response to Decision Letter 0]

17 Jan 2024

See attached response document.

---

## [Editor Report · Decision Letter 1]

5 Feb 2024

Reproductive coercion and abuse in intimate relationships: Women's perceptions of perpetrator motivations

PONE-D-23-29096R1

Dear Dr. Tarzia,

We’re pleased to inform you that your manuscript has been judged scientifically suitable for publication and will be formally accepted for publication once it meets all outstanding technical requirements.

Kind regards,

Michal Mahat-Shamir, Ph.D.

Academic Editor

PLOS ONE
---

## [Editor Report · Acceptance letter]

12 Feb 2024

PONE-D-23-29096R1 

PLOS ONE

Dear Dr. Tarzia, 

I'm pleased to inform you that your manuscript has been deemed suitable for publication in PLOS ONE. Congratulations! Your manuscript is now being handed over to our production team.

Kind regards, 

on behalf of

Prof. Michal Mahat-Shamir 

Academic Editor

PLOS ONE